# The Development of a Prediction Model Based on Random Survival Forest for the Postoperative Prognosis of Pancreatic Cancer: A SEER-Based Study

**DOI:** 10.3390/cancers14194667

**Published:** 2022-09-25

**Authors:** Jiaxi Lin, Minyue Yin, Lu Liu, Jingwen Gao, Chenyan Yu, Xiaolin Liu, Chunfang Xu, Jinzhou Zhu

**Affiliations:** 1Department of Gastroenterology, The First Affiliated Hospital of Soochow University, Suzhou 215000, China; 2Suzhou Clinical Center of Digestive Diseases, Suzhou 215000, China

**Keywords:** pancreatic cancer, random survival forest, surgery, machine learning, visualization, the Surveillance, Epidemiology, and End Results Program (SEER)

## Abstract

**Simple Summary:**

Surgery is the main treatment to cure pancreatic cancer (PC). However, the 5-year survival rate of surgical resection is only 10–20%. The aim of our study was to develop a prediction model with the novel machine learning algorithm random survival forest (RSF) and to offer easy-to-use prediction tools, including risk stratification and individual prognosis. The study would benefit patients and physicians in postoperative management and facilitate personalized medicine.

**Abstract:**

Accurate prediction for the prognosis of patients with pancreatic cancer (PC) is a emerge task nowadays. We aimed to develop survival models for postoperative PC patients, based on a novel algorithm, random survival forest (RSF), traditional Cox regression and neural networks (Deepsurv), using the Surveillance, Epidemiology, and End Results Program (SEER) database. A total of 3988 patients were included in this study. Eight clinicopathological features were selected using least absolute shrinkage and selection operator (LASSO) regression analysis and were utilized to develop the RSF model. The model was evaluated based on three dimensions: discrimination, calibration, and clinical benefit. It found that the RSF model predicted the cancer-specific survival (CSS) of the postoperative PC patients with a c-index of 0.723, which was higher than the models built by Cox regression (0.670) and Deepsurv (0.700). The Brier scores at 1, 3, and 5 years (0.188, 0.177, and 0.131) of the RSF model demonstrated the model’s favorable calibration and the decision curve analysis illustrated the model’s value of clinical implement. Moreover, the roles of the key variables were visualized in the Shapley Additive Explanations plotting. Lastly, the prediction model demonstrates value in risk stratification and individual prognosis. In this study, a high-performance prediction model for PC postoperative prognosis was developed, based on RSF The model presented significant strengths in the risk stratification and individual prognosis prediction.

## 1. Introduction

Pancreatic cancer (PC) is a severe malignant tumor, with the characteristic of poor prognosis and remarkable aggressiveness [1]. PC was one of five causes of death from cancer, of which the relative 5-year survival rate is only 8.2% [2,3]. The population of PC diagnosed has doubled in the past twenty years, which may be associated with the aging population, smoking, obesity, alcohol usage, diabetes, etc. [4,5,6,7,8,9]. The therapies, such as immunotherapy, chemotherapy, and radiotherapy, were less satisfactory in the field of PC [10,11]. Surgical resection is still now the only hope for a cure of PC [12]. With the assistance of adjuvant treatment, such as gemcitabine or FOLFIRINOX (fluorouracil + leucovorin + irinotecan + oxaliplatin), the 5-year survival rate of PC patients who had undergone complete macroscopic resection is up to 19–26.1% [13]. Meanwhile, for borderline resectable pancreatic cancer, neoadjuvant chemoradiotherapy combined with surgery also improves the 5-year overall survival rate up to 20.5% [14].

In this case, a practicable postoperative prognosis model for PC could benefit the clinical management of PC. A few clinical prediction models for survival after surgery for PC have been developed. Xu et al. constructed the overall survival of the pancreatic ductal adenocarcinoma model (C-index = 0.887), with 265 patients using the multivariable Cox regression analysis [15]. Likewise, Tol et al. developed the postoperative survival model (C-index = 0.658) with 350 patients [16]. However, the insufficiency of sample size and the homogeneity of prediction algorithms limit the performance of these models.

Recently, a novel machine learning algorithm, random survival forest (RSF), has been proposed to predict disease progression. With the characteristics of high performance and interpretability, the RSF is gaining ground. Until now, there have been no reports about the application of RSF in the postoperative prognosis of PC.

In this study, we aimed to use the data from the Surveillance, Epidemiology, and End Results Program (SEER) database, and to propose the postoperative prognostic model of PC based on the RSF algorithm. We also attempted to provide physicians and patients with prediction tools to assess the risk stratification and individual prognosis of postoperative of PC.

## 2. Materials and Methods

### 2.1. Study Population

The data were extracted from the SEER Researcher Plus Database (Nov 2020 Sub). The criteria for inclusion were: (1) the site and morphology code were ‘Pancreas’; (2) the histological codes included 8010–8049 (epithelial neoplasms), 8140–8389 (adenomas and adenocarcinomas), 8440–8499 (cystic, mucinous and serous), 8500–8548 (Ductal and lobular neoplasms), 8560–8679 (complex epithelial neoplasms) (the International Classification of Tumor Diseases Third Edition (ICD-O-3). (3) the behavior recode for analysis was malignant. The exclusion criteria were: (1) patients younger than 18; (2) the cause of death or follow-up survival months flag was unavailable; (3) patients lacking clinical information, such as surgical information, primary site codes, T stage (AJCC 7th), N stage (AJCC 7th), or clinical grade. The endpoint of the study was CSS (Cancer-Specific Survival), the duration interval between the first diagnosis and the death from pancreatic cancer.

### 2.2. Variables

The following variables were collected: the demographic information, including age, race, sex, and marital status; the clinical features, such as histological type, AJCC stage (AJCC 7th), T stage, N stage, M stage, the site of the tumor, tumor size, and clinical grade were incorporated. Meanwhile, the variables regarding the surgery, involving the surgery, the number of examined lymph nodes, the number of positive lymph nodes, and the rate of positive lymph nodes were retained. The records concerned chemotherapy and radiation were also included in the study.

### 2.3. The Development of Models

Patients were randomly divided into the training set, the validation set, and the test set at a ratio of 7:1:2. Variable selection was the main step before the modelling. The least absolute shrinkage and selection operator (LASSO) regression analysis was adopted in the RSF model. Then, the hyperparameters among the RSF model were fine-tuned in the validation set using the grid search method, which let us select the best hyperparameters from a list of options that we provide. The hyperparameters and range of grid search are below: the number of estimators (10, 100, 500, 1000); minimum of samples split (1, 3, 5, 10, 15, 20); minimum of samples leaf (1, 3, 4, 10, 15, 20). Two different algorithms containing Cox regression analysis and neural networks were also applied to develop the postoperative prognostic model for comparison. The Cox regression model was based on the multivariable Cox regression analysis, in which the variables were selected by the univariable regression analysis. The Deepsurv model was developed based on neural network algorithms. The integrated variables were directly loaded into the Deepsurv model without selection. The grid search is also applied in the Deepsurv model to look for the best parameters. Figure 1 illustrates the flowchart of the models’ development.

### 2.4. The Evaluation and Interpretation of Models

The models’ performance was evaluated in the test set. The following evaluation metrics were adopted: the concordance index (C-index) and the area under the operating characteristic curve (1, 3, and 5 years). The model’s calibration was evaluated by the Brier score. When Brier score ≤ 0.25, the model was considered to have favorable calibration. The decision curve analysis (DCA) was applied to calculate the clinical net benefit of the model [17].

The interpretation predictive model was essential for supporting medical decision-making, in which physicians could simply understand how the models make the prediction regarding the postoperative prognosis in a transparent manner. The Shapley Additive Explanations (SHAP) plot, which was a game-theoretic approach to explaining the output of model, demonstrated the contribution of the variables to the outcome.

### 2.5. The RSF Risk Stratification of Patients

The RSF risk stratification was based on the risk score, which was computed by the expected number of events for particular terminal node in the RSF model, could quantify patients’ postoperative hazards. Using X-tile software, we stratified patients according to their risk scores. The RSF risk stratification was tested by the Kaplan-Meier curve survival analysis and checked by log-rank test. To verify the value of the clinical implement, the survival predictions for patients at 1, 3, and 5 years after surgery of the RSF risk stratification with the traditional AJCC stage were also compared.

### 2.6. The Individual Prediction

The individual visual prediction was composed of the survival probability plot and local SHAP plot, which offer explicit individual prediction from the perspective of survival expectation and risk factors. The survival probability was calculated by the instance in each terminal using the non-parametrically estimate. The local SHAP plot is the local interpretation form of SHAP plot, which shows the contribution of variables for a given instance. According to the plot, the linkages between risk factors and the individual prognosis were established.

### 2.7. Statistical Analysis

The difference between demographic and clinical information was evaluated using the Wilcoxon test for continuous variables, while the χ2 test or Fisher’s exact test were used for categorical variables between the training set and the validation set. Two-tailed *p* values less than 0.05 were believed to be statistically significant. Python (Version 3.8, Van Rossum, Scotts Valley, CA, USA) was implemented to derive the models. The Cox model and RSF model were based on the scikit-survival module (Version 0.17.2, Sebastian P), while the Deepsurv model relied on the Pycox module. The fundamental data analysis was conducted by R (Version 4.1.2, RCoreTeam, Vienna, Austria).

## 3. Results

### 3.1. The Characteristics of Patients

A total of 3988 patients were included in the study. The demographic and clinical information of these patients between the training set and the test set was summarized in Table 1.

### 3.2. The Development of Models

RSF model: Using the LASSO regression analysis, eight variables were selected for the RSF model with the minimum criteria: age, histologic type, AJCC stage, T stage, N stage, clinical stage, the number of positive lymph nodes, and the rate of positive lymph nodes. The procedures for selecting variables were shown in Figure 2. Taking advantage of the grid search, the optimal structure of the RSF model comprised 500 estimators, 10 minimum of samples split, and 10 minimum of samples leaf.

Cox model: According to the univariable Cox regression analysis, the significant variables contained age, chemotherapy, histological type, AJCC stage, surgery, T stage, N stage, the primary site of the tumor, clinical grade, tumor size, the number of positive lymph nodes, and the rate of positive lymph nodes. In the multivariable Cox regression analysis, age, chemotherapy, AJCC stage, T stage, the primary site of the tumor, tumor size, and the rate of positive lymph nodes were identified as prognostic factors for postoperative pancreatic cancer. The details of the multivariable Cox regression analysis are shown in Appendix A.

Deepsurv model: After grid search, the backbone of the neural networks contains three layers, and the nodes were 10, 5, and 5 from the top down (Appendix A). The details are shown in the Deepsurv model section of the Appendix A.

### 3.3. The Evaluation and Interpretation of the Models

The models’ performance was checked in the test set. Models’ performance is shown in Table 2. The Brier scores of all models were less than 0.25, which demonstrated their good calibration. The RSF model outperformed other models as its highest C-index (0.723). According to Figure 3, the DCA regarding the RSF model showed fair clinical net benefits in 1, 3, and 5 years.

Furthermore, the RSF model was interpreted visually. In the SHAP plot (Figure 4), the variables in the model were listed in descending order of importance. The positive lymph node rate was considered the most significant variable, followed by clinical grade, age, the positive lymph nodes, histological type, etc. Meanwhile, the survival analysis of the categorical risk variables is supplied in Appendix A.

### 3.4. The RSF Risk Stratification of Patients

The stratification of the patients was of major importance for guiding patient management. Patients were divided into the high-risk group (risk score > 157), medium-risk group (123 ≤ risk score ≤ 157), and the low-risk group (risk score < 123) with the assistance of X-tile (Appendix A). The specific process was illustrated by the section of the optimal cut-off values of RSF risk stratification proposed in the Appendix A. The results of the Kaplan-Meier analysis and log-rank test between the high-risk group, medium-risk, and low-risk group were presented in Figure 5, which demonstrated significant difference between three groups. The medium survival time of different RSF stratifications are shown in Table 3 and Table 4. Meanwhile, as shown in Table 5, the RSF risk stratification demonstrated better discrimination in 1, 3, and 5 years after the PC surgery by comparison with traditional AJCC stage.

### 3.5. The Individual Postoperative Prognostic Prediction

Three patients were chosen at random for the individual postoperative prognostic prediction demonstration. Figure 6A shows the individual prediction survival fraction. The local SHAP explained the prognosis of each patient from the point of variables’ contribution, which are shown in Figure 6B–D.

**Patient #1:** 70 years male, AJCC stage was IV, T stage was III, N stage was N1, M stage was M1. The clinical grade was II. The positive lymph nodes were 13. The positive lymph nodes rate was 0.54. The CSS time was 10 months.

**Patient #2:** 74 years female, AJCC stage was II, T stage was III, N stage was N0, M stage was M0. The clinical grade was III. The positive lymph nodes were 0. The positive lymph nodes rate was 0. The CSS time was 30 months.

**Patient #3:** 53 years male, AJCC stage was I, T stage was II, N stage was N0, M stage was M0. The clinical grade was I. The positive lymph nodes were 0. The positive lymph node rate was 0. The CSS time was 139 months.

The RSF model and the prediction tools are available in the website: https://github.com/Lin725/RSF-model (accessed on 16 August 2022). It provides a quicker and more intuitive way of predicting. Meanwhile, the clinical practitioners can furthermore optimize our model in the future on the basis of the open-source codes.

## 4. Discussion

This study reported the postoperative prognostic model of PC using the SEER database based on the RSF algorithm. The RSF model demonstrated better calibration and discrimination in predicting 1-, 3-, and 5-year CSS of postoperative PC patients than the models by Cox regression and neural networks. Through the visual interpretation for the model, the positive lymph node rate was identified as the most significant risk variable, followed by clinical grade, age, and positive lymph nodes. Furthermore, the risk stratification and individual postoperative prognostic prediction based on the RSF model showed potential in the clinical practice.

The RSF algorithm, which was first proposed in 2008, has emerged as an intuitive tool for predicting the prognosis [18]. Compared to the traditional Cox regression analysis, the RSF algorithm tends to develop better performing models, particularly when processing the high-dimensional data [19]. Meanwhile, due to the restriction of the proportional hazard assumption, the application of Cox regression analysis was limited. However, there was no such restriction in the RSF, benefiting from the non-parametric structure. Despite the fact that the models developed by the neural networks always show impressive performance, the characteristic of the “black box” remained an obstacle [20]. The RSF algorithm could obtain the balance between the model fitting and the interpretation, as shown in this study.

The analysis of risk factors, which were identified in the RSF prediction model might facilitate the surgery management and reduce the medical burden. The rate of positive lymph nodes had a decisive influence on the patients’ outcome. With the increasing rate of positive lymph nodes, the survival time of patients was reduced significantly. Previous studies also noted the unfavorable impact of increasing rate of the positive lymph nodes [16,21,22,23]. However, the impact of specific nodes rate on the outcome was inconclusive.

The clinical grade described how abnormal the cancer cells and tissue look under a microscope when compared to healthy cells. In our study, the high clinical grade led to the poor prognosis of postoperative patients. The mechanism behind the phenomenon might be due to the more remarkable aggression of the high clinical grade cell compared to the low clinical grade cells. Yang et al. and Geer et al. also demonstrated that the prognosis of PC with a higher clinical grade was less satisfactory [24,25].

We found the differences in histological types also reflected in the postoperative prognosis, which were also referred to in prior study [26,27]. The epithelial neoplasms and cystic, mucinous and serous neoplasms seemed to gain more benefits from the surgery in our study. Pokrywa et al. also pointed that the median overall survival of the cystic mucinous neoplasm was 52.6 months, while the ductal adenocarcinoma was only 20.2 months. However, the uneven distribution of histological types may lead to a biased conclusion. In this study, the adenocarcinomas counted for 51%, while the cystic mucinous neoplasm only counted for 4.6%. More information regarding the rare histological type needs to be collected in the future to reach more solid conclusions.

The AJCC stage, consisting of the T stage, N stage, and M stage, was the basic tumor staging method, which could make a rough assessment for patients’ prognosis [28]. The median overall survival of stage I was 51 months, while it was 19 months in stage II, III, and IV in our study. The high AJCC stage was the sign of an unsatisfactory outcome. Meanwhile, the T stage and N stage, which indicated the extent of tumor invasion and the lymph node metastasis also significantly affected the patients’ outcome. Considering that patients who had distant metastases usually did not undergo surgical treatment, M stage was not significant in our model.

Notably, the elderly patients indicated a low survival rate. This phenomenon might be related to the immune deficiency and physical deterioration of elderly patients [29]. For the patients with the above-mentioned risk factors, the physicians should emphasis more attention to their postoperative prognosis.

The RSF risk stratification could make assessment of the patient’s prognosis based on the intraoperative situation. Compared to well-established score systems, such as Heidelberg-Score, PANAMA-score, we incorporated a larger sample size for development and validation (1071 patients for Heidelberg-Score, 216 patients for PANAMA-score, and 3988 patients for RSF risk stratification) [30,31]. The RSF risk stratification also made innovations in the field of algorithm. However, due to the heterogeneity of the variables incorporated in each score system, it was difficult for us to compare their performance fairly. It was worthwhile to compare each score system in a large population in the feature. When using the RSF risk stratification, physicians could evaluate the survival cycle of patients. Meanwhile, physicians should pay more attention to the high-risk group (risk score > 157). Because these patients had a median survival time of only 14 months. They were more likely to suffer from early death. The individual postoperative prognostic prediction provided a more specific view of patients’ prognosis. Compared to the nomogram, which had been widely used as prognostic device in oncology and medicine, the RSF model provided a more flexible approach to forecasting [32]. The disadvantage of the nomogram was that it can only predict the survival situation at the exact point in time. The lack of intuition regarding the impact of risk factors on individual outcomes was another shortcoming of the nomogram, whereas the RSF model improved the form of prediction. With an individual survival probability curve, patients’ postoperative prognosis was presented from a more precise perspective. Additionally, the local SHAP plot visually explained the impact of risk factors on individual survival outcomes. Our study attempted the new clinical prediction application and assessed its feasibility.

Some limitations of this study should be referred to. First, the retrospective nature of the study leads to potential selection bias. Second, the training set and the test set are extracted from the same database, which may reduce the model’s generalizability. The external test set with large patient numbers was required to future validate the model. Third, due to the restriction of the SEER database, some possible variables, such as the usage of drugs, genetic factors, etc., are not available. The incorporation of additional potential variables may enhance the performance of the RSF model. Meanwhile, the lack of further exploration of adjuvant therapies (adjuvant therapy and neoadjuvant therapy) is also a shortcoming of our study.

## 5. Conclusions

Using the RSF algorithm, we developed the high-performance prediction model regarding the postoperative prognosis of PC patients. Furthermore, we stratified the postoperative populations and predicted the individual prognosis comprehensively with the RSF model. We also provided physicians and patients with an easy-to-use prediction tool for postoperative management and facilitate personalized medicine. Our study supports that RSF algorithm shows promise in future clinical research and practice.

## Figures and Tables

**Figure 1 cancers-14-04667-f001:**
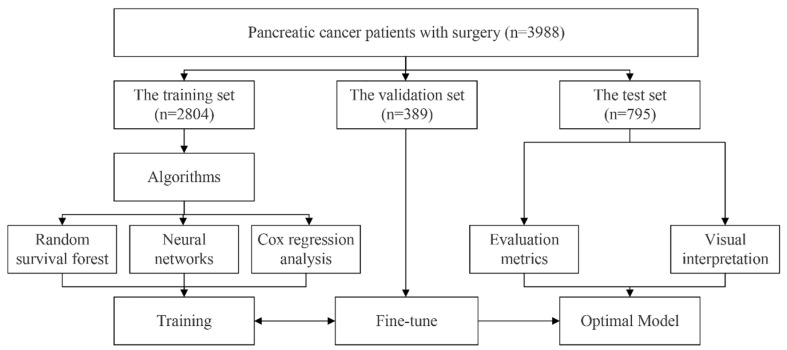
The flowchart of developing models.

**Figure 2 cancers-14-04667-f002:**
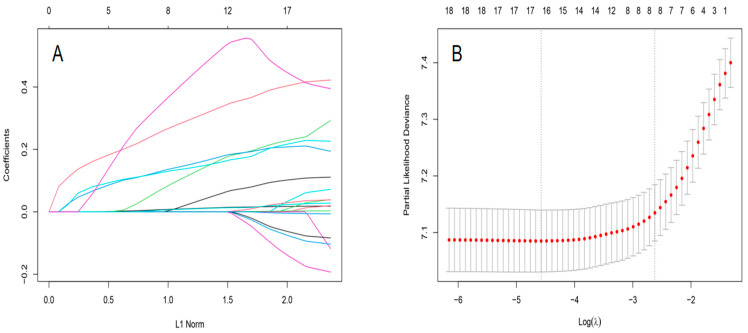
The results of LASSO regression analysis for RSF model. (**A**) LASSO coefficient profiles of the expression of 21 variables. (**B**) Selection of the λ in the LASSO regression analysis via 10-fold cross-validation. The dotted vertical lines are plotted at the optimal values following the minimum criteria (right) and “one standard error” criteria (left).

**Figure 3 cancers-14-04667-f003:**
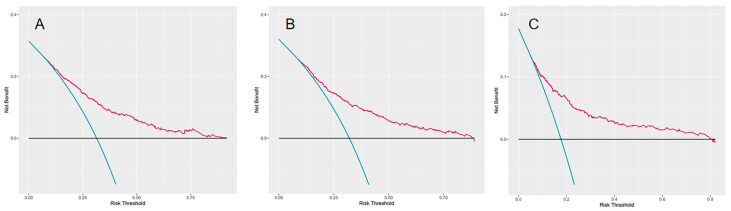
The decision analysis curves of RSF model. (**A**) The one-year decision analysis curve of RSF model. (**B**) The three-year decision analysis curve of RSF model. (**C**) The five-year decision analysis curve of RSF model. In the decision analysis curve, the *x*-axis represented the threshold probability while the *y*-axis represented the clinical net benefits. The blue line in the DCA plot reflects the strategy of “assume all patients have received the assessment of the RSF model”, while the horizontal black line demonstrates the strategy of “assume no patient has received the assessment of the RSF model”.

**Figure 4 cancers-14-04667-f004:**
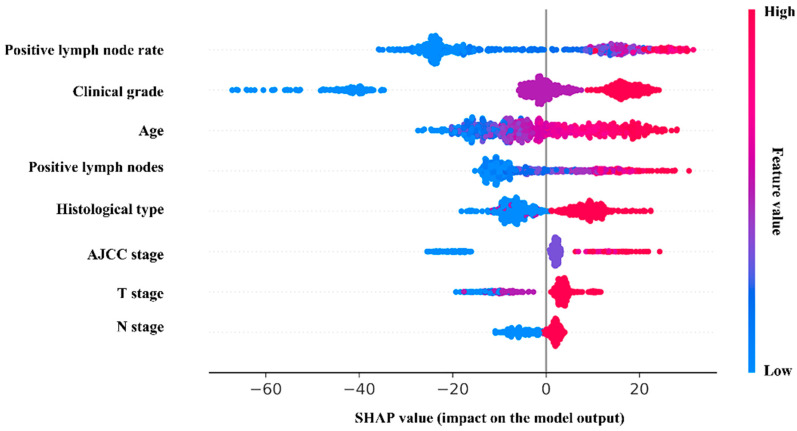
The SHAP plot of the RSF model. In the SHAP plot, the length of the horizontal axis where each variable is located represents the variable’s contribution to the outcome. The color of the dot symbolized the numerical value of the variable. For example, the variable (positive lymph node rate) is the most significant risk factor. The higher the rate is, the higher the probability of poor prognosis is.

**Figure 5 cancers-14-04667-f005:**
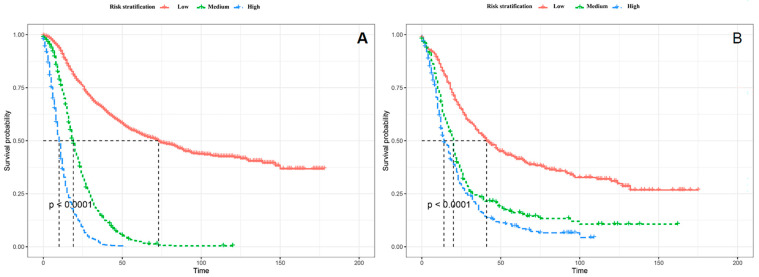
The RSF risk stratification patients. (**A**). The RSF risk stratification of patients in the training set. (**B**). The RSF risk stratification of patients in the test set.

**Figure 6 cancers-14-04667-f006:**
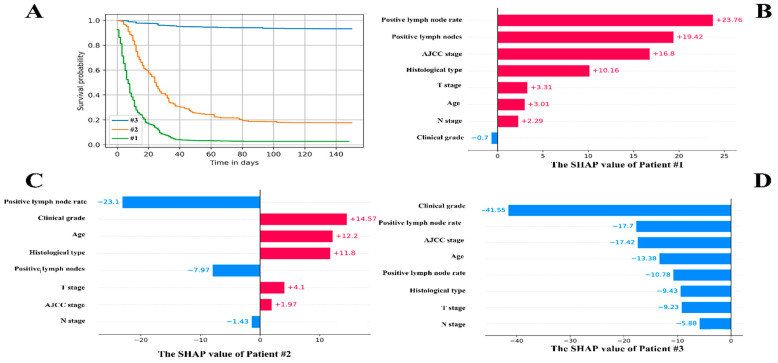
The individual postoperative prognostic prediction. (**A**). The estimated survival function of patients. The green line symbolizes the patients A while the yellow line represents the patient B. The blue line on behalf of the patient C. (**B**). The local SHAP plot of the patient #1. (**C**). The local SHAP plot of the patient #2. (**D**). The local SHAP plot of the patient #3. The red ribbon in the local SHAP plot represented the risk factors, which promoted the poor prognosis, whereas the blue ribbon was the relatively protective factors.

**Table 1 cancers-14-04667-t001:** The information for postoperative pancreatic cancer patients in the training set and the test set.

Characteristic	Training Set (*n* = 2804)	Test Set (*n* = 795)	*p* Value
Age	65 (58, 73)	65 (57, 72)	0.5
Race			0.9
White	2215 (79%)	629 (79%)	
Black	293 (10%)	82 (10%)	
Asian or Pacific Islander	292 (10%)	84 (11%)	
other	4(0.1%)	0 (0%)	
Sex			0.5
Male	1432 (51%)	396 (50%)	
Female	1372 (49%)	399 (50%)	
Marital status			0.5
Married	2067 (74%)	577 (73%)	
Single	737 (26%)	218 (27%)	
Radiation			0.4
Yes	998 (36%)	271 (34%)	
No	1806 (64%)	524 (64%)	
Chemotherapy			0.2
Yes	1862 (66%)	508 (64%)	
No	942 (34%)	287 (36%)	
Histological type			0.5
Epithelial neoplasms	56 (2.0%)	9 (1.1%)	
Adenomas and adenocarcinomas	1435 (51%)	413 (52%)	
Cystic, mucinous, and serous	130 (4.6%)	32 (4.0%)	
Ductal and lobular neoplasms	1144 (41%)	332 (42%)	
Complex epithelial neoplasms	39 (1.4%)	9 (1.1%)	
Surgery			0.1
Local excision	5 (0.2%)	5 (0.6%)	
Partial pancreatectomy	464 (17%)	118 (15%)	
Local or partial pancreatectomy and duodenectomy	1882 (67%)	537 (68%)	
Total pancreatectomy	79 (2.8%)	32 (4.0%)	
Total pancreatectomy and subtotal gastrectomy or duodenectomy	221 (7.9%)	68 (8.6%)	
Extended pancreatoduodenectomy	132 (4.7%)	33 (4.2%)	
Pancreatectomy	21 (0.7%)	2 (0.3%)	
AJCC stage			>0.9
I	357 (13%)	103 (13%)	
II	2185 (78%)	621 (78%)	
III	102 (3.6%)	26 (3.3%)	
IV	160 (5.7%)	45 (5.7%)	
T stage			0.7
T1	204 (7.3%)	55 (6.9%)	
T2	449 (16%)	115 (14%)	
T3	2039 (73%)	594 (75%)	
T4	112 (4.0%)	31 (3.9%)	
N stage			0.1
N0	942 (34%)	294 (37%)	
N1	1862 (66%)	501 (63%)	
M stage			>0.9
M0	2644 (94.3%)	750 (94.3%)	
M1	160 (5.7%)	45 (5.7%)	
Site			0.4
Pancreas Head	1969 (70%)	572 (72%)	
Pancreas Body Tail	566 (20%)	158 (20%)	
Other	269 (9.6%)	65 (8.2%)	
Clinical grade			0.2
I	449 (16%)	108 (14%)	
II	1315 (47%)	394 (50%)	
III	985 (35%)	283 (36%)	
IV	55 (2.0%)	10 (1.3%)	
Tumor size (mm)	32 (25, 45)	32 (25, 42)	0.5
Examined lymph nodes	14 (9, 21)	14 (9, 21)	0.6
Positive lymph nodes	1 (0, 3)	1 (0, 4)	0.2
Positive lymph nodes rate (%)	0.10 (0.00, 0.25)	0.08 (0.00, 0.25)	0.2

Notes: AJCC: American Joint Committee on Cancer

**Table 2 cancers-14-04667-t002:** The models’ performance in the test set.

Model	AUC			Brier Score			C-Index
	1-Year	3-Year	5-Year	1-Year	3-Year	5-Year	
RSF model	0.753	0.744	0.759	0.188	0.177	0.131	0.723
Cox model	0.736	0.737	0.76	0.193	0.181	0.132	0.670
Deepsurv model	0.744	0.742	0.749	0.202	0.175	0.122	0.700

**Table 3 cancers-14-04667-t003:** The medium survival time (months) of different risk stratifications in the training set.

RSF Risk Stratification	Number	Events	Median	0.95 LCL	0.95 UCL
Low-risk	1332	644	73	63	85
Medium-risk	766	719	19	18	20
High-risk	706	686	10	9	11

Note: LCL: Low confidence interval, UCL: Up confidence interval.

**Table 4 cancers-14-04667-t004:** The medium survival time (months) of different risk stratifications in the test set.

RSF Risk Stratification	Number	Events	Median	0.95 LCL	0.95 UCL
Low-risk	218	41	41	35	53
Medium-risk	156	20	20	17	21
High-risk	212	14	14	12	18

Note: LCL: Low confidence interval, UCL: Up confidence interval.

**Table 5 cancers-14-04667-t005:** The area under the curve of RSF risk stratification vs. AJCC Stage.

	1-Year	3-Year	5-Year
RSF risk stratification	0.667	0.693	0.688
AJCC stage	0.568	0.603	0.622
*p* value	<0.001	<0.001	0.012

## Data Availability

The data used in this study are available from the Surveillance, Epidemiology, and End Results Program (SEER) database (Nov 2020 Sub).

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
