# Peer review of "The Development of a Prediction Model Based on Random Survival Forest for the Postoperative Prognosis of Pancreatic Cancer: A SEER-Based Study"

_cancers, 2022, doi:10.3390/cancers14194667_

Round 1
Reviewer 1 Report
Dear Dr. Lin and Colleagues,
I had the pleasure in reading your manuscript The Development of a Prediction Model Based on Random Survival Forest For the Post-Operative Prognosis of Pancreatic cancer: A SEER-Based Study.
I have a few concerns
1. Line 19 Abstract: Change grammar.
Best wishes
Author Response
Response to the Reviewer Comments
We would like to thank you for your efforts in reviewing our manuscript titled The development of a prediction model based on random survival forest for the postoperative prognosis of pancreatic cancer: A SEER-Based study, and providing the helpful comment, which will all prove invaluable in the revision and improvement of our paper, as well as in guiding our research in the future.
We have studied your comments point by point, revised the manuscript accordingly. The amendment is highlighted in yellow in the revised manuscript. All authors have approved the response letter and the revised version of the manuscript. According to your suggestions, we have made the following revisions on this manuscript.
[1] Line 19 Abstract: Change grammar.
Response: We are very sorry for our incorrect writing and the grammatical errors were rectified at Line 19. We have made correction according to your comment.
Once again, thank you very much for your constructive comment which would help us both in English and in depth to improve the quality of the paper.
Reviewer 2 Report
I read with great interest the paper by Lin et al. on a new prediction model to stratify the survival after surgery for PDAC. Stratification for PDAC is certainly a major step for individualized medicine in the future and, therefore, novel tools which are easily to implement in clinical routine are urgently needed. The authors developed a post-resection model by using random survival forest in a SEER database derived patient cohort. The authors identified in the training cohort 8 variables which were used for the model (Clinical and pathological determinants). The AUC was above 0.72 and performed better in contrast to Cox and Deepsur models. However, I have some major critics regarding this paper.
First, I think the postresection survival in the past two decades became much better than the authors report in their introduction. I would recommend citing some up-to-date trials on outcome with modern (neo-)adjuvant regimes (e.g. FOLFIRINOX) where the 5-year survival is certainly above 10%.
Second, I have trouble with the patient cohort derived from the SEER database. It is just mixing all patients with a pancreatic neoplasm together in one basket. I assume there are also patients with low- or high-grade dysplasia included in the category “Cystic, mucinous and serous” which is somehow conflicting in a study on pancreatic cancer. What about patients with M1 disease which reflects certainly a different stage and biology of PDAC.
Third, I think it is not timely to include patients which are only categorized according to the 7th AJCC edition which was already updated 2017/18.
Fourth, the others do not discriminate at all between patients who received neoadjuvant or adjuvant therapy. Since there is an increasing number of patients who receive NAT for borderline or locally advanced PDAC a discrimination of the model should be applied.
Lastly, I think the authors should invest more time to discuss their novel post-resection score in contrast to well-established scores (MSKCC), Heidelberg-Score (Hartwig et al. 2011) as well as in the era of neoadjuvant therapy (PANAMA-score). What are benefits of this score. What do the authors recommend for patients with a high- or low RSF score. Especially the recommendation for or against surgery (line 296) shouldn’t be based on a post-resection score which is proposed in the current study.
Author Response
We would like to thank you for your efforts in reviewing our manuscript.
Please see the attachment.

Reviewer 3 Report
Comments and Suggestions for Authors
The article by Lin et al. focuses on a relevant topic in oncology and describes an example of developing a predictive model for medicine. The use of computer technology meets the current stage of development of health care systems in modern countries and is in great demand. This is especially important for improving the long-term outcomes of severe diseases such as pancreatic cancer.
In the presented manuscript, the authors describe the development, testing and comparing the performance of various predictive models that can stratify patients that underwent surgical treatment for pancreatic cancer into groups with high, medium, and low median survival.
The authors provide a sufficient introduction and do a good job of describing the origin and manipulation of the SEER data required to develop a predictive model. Figures, tables and methodology are also described clearly and concisely. The data fully support the authors' conclusions that random survival forest is as good or even better approach to building a predictive model for pancreatic cancer patients than neural networks or widely used Cox regression.
Separately, I would like to thank the authors for sharing the source code of the model and the opportunity to test it on clinical data.
The manuscript is certainly well written and attractive, and I recommend accepting it for publication with a slight correction regarding Figure 6 and its legend.
Minor issues:
On Fig. 6 panel labels are hard to read, especially panel D where the 'D' is above the data. I would recommend labeling panels outside of the data area for all figures in the manuscript.
Line 233: In the legend of Figure 6, panel D is labeled as C.
Round 2
Reviewer 2 Report
The authors addressed the comments sufficiently.